# Learning from the Experience of Others: Approximate Empirical Bayes in Neural Networks

## Abstract

Learning deep neural networks could be understood as the combination of representation learning and learning halfspaces. While most previous work aims to diversify representation learning by data augmentations and regularizations, we explore the opposite direction through the lens of empirical Bayes method. Specifically, we propose a matrix-variate normal prior whose covariance matrix has a Kronecker product structure to capture the correlations in learning different neurons through backpropagation. The prior encourages neurons to learn from the experience of others, hence it provides an effective regularization when training large networks on small datasets. To optimize the model, we design an efficient block coordinate descent algorithm with analytic solutions. Empirically, we show that the proposed method helps the network converge to better local optima that also generalize better, and we verify the effectiveness of the approach on both multiclass classification and multitask regression problems with various network structures.

## 1 Introduction

Empirical Bayes methods provide us a powerful tool to obtain Bayesian estimators even if we do not have complete information about prior distribution. The literature on the empirical Bayes methods and their applications are abundant (Stein, 1956; Robbins, 1956; James & Stein, 1961; Efron & Morris, 1973; 1977; Efron et al., 2001; Carlin & Louis, 2010; Efron, 2012). Existing studies on parametric empirical Bayes methods focus on the setting where the likelihood function and the prior are assumed to have specific forms, e.g., exponential family distribution and its conjugate prior, so that marginal distribution of data has a closed form from which an estimator of the hyperparameter in the prior distribution can be obtained. While such assumption helps to simplify the setting in order to demonstrate the power of the method, it restricts us from using more expressive and rich models.

In this paper we explore extending the empirical Bayes method to expressive nonlinear models using deep neural networks. Although deep neural networks have been widely applied in various domains (Krizhevsky & Hinton, 2009; LeCun et al., 2015; He et al., 2016), usually its parameters are learned via the principle of maximum likelihood, hence its success crucially hinges on the availability of large scale datasets. On the other hand, Bayesian modeling lends us a powerful and principled tool to prevent overfitting by incorporating prior knowledge into the design of prior distribution. To this end, we propose a regularization approach for the weight matrix in neural networks through the lens of the empirical Bayes method. We aim to address the problem of overfitting when training large networks on small dataset. Our key insight stems from the famous argument by Efron (2012): *It is beneficial to learn from the experience of others.* Specifically, from an algorithmic perspective, we argue that the connection weights of neurons in the same layer (row/column vectors of the weight matrix) will be correlated with each other through the backpropagation learning. Hence by learning from other neurons in the same layer, a neuron can "borrow statistical strength" from other neurons.

As an illustrating example, consider a simple setting where the input $\mathbf{x} \in \mathbb{R}^d$ is fully connected to a hidden layer $\mathbf{h} \in \mathbb{R}^p$, which is further fully connected to the single output $\hat{y} \in \mathbb{R}$. Let $\sigma(\cdot)$ be the nonlinear activation function, e.g., ReLU (Nair & Hinton, 2010), $W \in \mathbb{R}^{p \times d}$ be the connection matrix between the input layer and the hidden layer, and $\mathbf{a} \in \mathbb{R}^p$ be the vector connecting the output and the hidden layer. Without loss of generality, ignoring the bias term in each layer, we have: $\hat{y} = \mathbf{a}^T \mathbf{h}, \mathbf{h} = \sigma(W\mathbf{x})$. Consider using the usual $\ell_2$ loss function $\ell(\hat{y}, y) = \frac{1}{2}|\hat{y} - y|^2$ and take the derivative of $\ell(\hat{y}, y)$ w.r.t. $W$. We obtain the update formula in backpropagation as $W \leftarrow W - \alpha(\hat{y} - y)(\mathbf{a} \circ \mathbf{h}') \mathbf{x}^T$, where $\mathbf{h}'$ is the componentwise derivative of $\mathbf{h}$ w.r.t. its input

argument, and $\alpha > 0$ is the learning rate. Realize that $(\mathbf{a} \circ \mathbf{h}') \, \mathbf{x}^T$ is a rank 1 matrix, and the component of $\mathbf{h}'$ is either 0 or 1. Hence the update for each row vector of $W$ is linearly proportional to $\mathbf{x}$. Note that the observation holds for any input pair $(\mathbf{x}, y)$, so the update formula implies that the row vectors of $W$ are correlated with each other. The above observation leads us to the following question: *can we define a prior distribution over $W$ that captures the correlations between its row vectors and column vectors?*

**Our Contributions**. To answer the above question, we develop an approximate empirical Bayes (AEB) framework to learn deep neural networks. Motivated by the example above, we propose a matrix-variate normal prior whose covariance matrix has a Kronecker product structure to capture the correlations between different neurons. The prior encourages neurons to learn from the experience of others, hence it provides an effective regularization when training networks on small datasets. Using tools from convex analysis, we design an efficient block coordinate descent algorithm with analytic solutions to optimize the model. Empirically, we show that the proposed method helps the network converge to better local optima that also generalize better, and we verify the effectiveness of the approach on both multiclass classification and multitask regression problems with various network structures.

## 2 PRELIMINARY

We first introduce the notation used throughout the paper and then give a brief discussion on the empirical Bayes method (Bernardo & Smith, 2001; Gelman et al., 2013; Efron & Hastie, 2016).

### 2.1 NOTATION AND SETUP

We use lowercase letter to represent scalar and lowercase bold letter to denote vector. Capital letter, e.g., $X$, is reserved for matrix. Calligraphic letter, such as $\mathcal{D}$, is used to denote set. We write $\text{Tr}(A)$ as the trace of a matrix $A$, $\det(A)$ as the determinant of $A$ and $\text{vec}(A)$ as $A$'s vectorization by column. $[n]$ is used to represent the set $\{1, \ldots, n\}$ for any integer $n$. Other notations will be introduced whenever needed. Suppose we have access to a training set $\mathcal{D}$ of $n$ pairs of data instances $(\mathbf{x}_i, y_i), i \in [n]$. We consider the supervised learning setting where $\mathbf{x}_i \in \mathcal{X} \subseteq \mathbb{R}^d$ and $y_i \in \mathcal{Y}$. For a regression problem, $\mathcal{Y} = \mathbb{R}$; for a binary classification problem, $\mathcal{Y} = \{1, -1\}$. Let $p(y \mid \mathbf{x}, \mathbf{w})$ be the conditional distribution of $y$ given $\mathbf{x}$ with parameter $\mathbf{w}$. The parametric form of the conditional distribution is assumed be known. In this paper, we consider a Bayesian setting where the model parameter $\mathbf{w}$ is sampled from a prior distribution $p(\mathbf{w} \mid \theta)$ with hyperparameter $\theta$. On the other hand, given $\mathcal{D}$, the posterior distribution of $\mathbf{w}$ is denoted by $p(\mathbf{w} \mid \mathcal{D}, \theta)$. From a Bayesian perspective, given an unseen instance $\mathbf{x}$, the goal is to infer the predictive distribution: $p(y \mid \mathbf{x}, \mathcal{D}, \theta) = \int p(y \mid \mathbf{x}, \mathbf{w}) \cdot p(\mathbf{w} \mid \mathcal{D}, \theta) \, d\mathbf{w}$, from which we can compute the mean, or the median, or other statistic (depending on the choice of the loss function) as our estimator of the unseen target variable $y$.

### 2.2 THE EMPIRICAL BAYES METHOD

To compute the predictive distribution, we need access to the value of the hyperparameter $\theta$. However, complete information about the hyperparameter $\theta$ is usually not available in practice. To this end, empirical Bayes method (Robbins, 1956; Efron & Morris, 1973) proposes to estimate $\theta$ from the data directly using the marginal distribution:

$$\hat{\theta} = \arg\max_{\theta} \; p(\mathcal{D} \mid \theta) = \arg\max_{\theta} \int p(\mathcal{D} \mid \mathbf{w}) \cdot p(\mathbf{w} \mid \theta) \, d\mathbf{w}. \tag{1}$$

Under specific choice of the likelihood function $p(\mathbf{x}, y \mid \mathbf{w})$ and the prior distribution $p(\mathbf{w} \mid \theta)$, e.g., exponential family distribution and its corresponding conjugate prior, we can solve the integral in (1) in closed form to obtain an analytic solution of $\hat{\theta}$, which can be subsequently plugged into the prior distribution to obtain a Bayesian estimator for the model parameter $\mathbf{w}$.

At a high level, by learning the hyperparameter $\theta$ in the prior distribution directly from data, the empirical Bayes method provides us a principled and convenient way to obtain a Bayesian estimator of the model parameter $\mathbf{w}$. In fact, when both the prior and the likelihood functions are normal, it has

been formally shown that the empirical Bayes estimators, e.g., the James-Stein estimator (James & Stein, 1961) and the Efron-Morris estimator (Efron & Morris, 1977), dominate the classic maximum likelihood estimator (MLE) in terms of quadratic loss for every choice of the model parameter $\mathbf{w}$. At a colloquial level, the success of empirical Bayes estimators can be attributed to the effect of *"learning from the experience of others"* (Efron, 2012), which also makes it a powerful tool in multitask learning (Long et al., 2017; Zhao et al., 2017) and meta-learning (Grant et al., 2018).

## 3 LEARNING WITH APPROXIMATE EMPIRICAL BAYES

### 3.1 APPROXIMATE EMPIRICAL BAYES

When the likelihood function $p(\mathcal{D} \mid \mathbf{w})$ is implemented as a neural network, the marginalization in (1) over model parameter $\mathbf{w}$ cannot be computed exactly. Nevertheless, instead of performing expensive Monte-Carlo simulation, we can use point estimate of $\mathbf{w}$ to approximate the integral as follows:

$$\int p(\mathcal{D} \mid \mathbf{w}) \cdot p(\mathbf{w} \mid \theta) \, d\mathbf{w} \approx p(\mathcal{D} \mid \hat{\mathbf{w}}) \cdot p(\hat{\mathbf{w}} \mid \theta), \tag{2}$$

where $\hat{\mathbf{w}} = \arg\max_{\mathbf{w}} p(\mathcal{D} \mid \mathbf{w}) \cdot p(\mathbf{w} \mid \theta)$ is the mode of the joint distribution. The above approximation is crude and will only be accurate if 1) the likelihood under $\hat{\mathbf{w}}$ dominates the likelihoods under other model parameters, or 2) for the fixed $\theta$, the prior distribution $p(\mathbf{w} \mid \theta)$ is sharply concentrated around its mode. When the size of the dataset $\mathcal{D}$ is large, the first condition is met according to the central limit theorem under some regularity conditions. We shall come back later to verify the validity of this approximation through numerical experiments in Sec. 4.

Given an estimate $\hat{\mathbf{w}}$, by maximizing the R.H.S. of (2) w.r.t. $\theta$, we can obtain $\hat{\theta}$ as an approximation of the maximum marginal likelihood estimator. As a result, we can use $\hat{\theta}$ to further refine the estimate $\hat{\mathbf{w}}$ by maximizing the posterior distribution as follows:

$$\hat{\mathbf{w}} \leftarrow \max_{\mathbf{w}} \ p(\mathbf{w} \mid \mathcal{D}) = \max_{\mathbf{w}} \ p(\mathcal{D} \mid \mathbf{w}) \cdot p(\mathbf{w} \mid \hat{\theta}). \tag{3}$$

The maximizer of (3) can in turn be used to better approximate the integral in (2). Formally, we can define the following optimization problem that characterizes our framework of the approximate empirical Bayes (AEB) method:

$$\max_{\mathbf{w}} \max_{\theta} \ \log p(\mathcal{D} \mid \mathbf{w}) + \log p(\mathbf{w} \mid \theta) \tag{4}$$

It is worth to connect the optimization problem (4) to the classic maximum a posteriori (MAP) inference and also discuss their difference. If we drop the inner optimization over the hyperparameter $\theta$ in the prior distribution. Then for any fixed value $\hat{\theta}$, (4) reduces to MAP with the prior defined by the specific choice of $\hat{\theta}$, and the maximizer $\hat{\mathbf{w}}$ corresponds to the mode of the posterior distribution given by $\hat{\theta}$. From this perspective, the optimization problem in (4) actually defines a series of MAP inference problems, and the sequence $\{\hat{\mathbf{w}}_j(\hat{\theta}_j)\}_j$ defines a solution path towards the final approximate empirical Bayes estimator. On the algorithmic side, the optimization problem (4) also suggests a natural block coordinate descent algorithm where we alternatively optimize over $\mathbf{w}$ and $\theta$ until the convergence of the objective function. An illustration of the framework is shown in Fig. 1. In next section, we give a specific prior distribution over the parameters of neural networks to capture the fact that neurons in the same layer are correlated with each other.

### 3.2 NEURAL NETWORK WITH MATRIX-NORMAL PRIOR

Inspired by the observation from Sec. 1, we propose to define a matrix-variate normal distribution (Gupta & Nagar, 2018) over the connection weight matrix $W$: $W \sim \mathcal{MN}(0_{p \times d}, \Sigma_r, \Sigma_c)$, where $\Sigma_r \in \mathbb{S}_{++}^p$ and $\Sigma_c \in \mathbb{S}_{++}^d$ are the row and column covariance matrices, respectively.[1] Equivalently, one can understand the matrix-variate normal distribution over $W$ as a multivariate normal distribution with a Kronecker product covariance structure over $\text{vec}(W)$: $\text{vec}(W) \sim \mathcal{N}(0_{p \times d}, \Sigma_c \otimes \Sigma_r)$. It

---

[1]The probability density function is given by $p(W \mid \Sigma_r, \Sigma_c) = \frac{\exp\left(-\operatorname{Tr}(\Sigma_r^{-1} W \Sigma_c^{-1} W^T)/2\right)}{(2\pi)^{pd/2} \det(\Sigma_r)^{d/2} \det(\Sigma_c)^{p/2}}.$

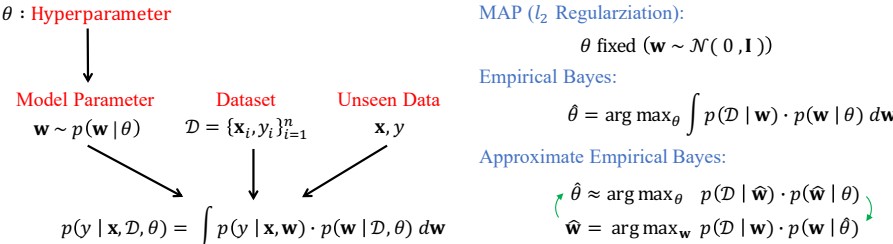

Figure 1: Illustration for Bayes/ Empirical Bayes, and Approximate Empirical Bayes.

is then easy to check that the marginal prior distributions over the row and column vectors of the weight matrix $W$ are given by:

$$W_{i:} \sim \mathcal{N}(\mathbf{0}_d, [\Sigma_r]_{ii} \cdot \Sigma_c), \quad W_{:j} \sim \mathcal{N}(\mathbf{0}_p, [\Sigma_c]_{jj} \cdot \Sigma_r).$$

We point out that the Kronecker product structure of the covariance matrix exactly captures our prior about the connection matrix $W$: the fan-in/fan-out of neurons in the same layer (row/column vectors of $W$) are correlated with the same correlation matrix in the prior, and they only differ at the scales (variances).

For illustration purpose, let us consider the simple feed-forward network discussed in Sec. 1. Consider a reparametrization of the model by defining $\Omega_r := \Sigma_r^{-1}$ and $\Omega_c := \Sigma_c^{-1}$ to be the corresponding precision matrices and plug in the prior distribution into the general approximate empirical Bayes framework (4). After routine algebraic simplifications, we reach the following concrete optimization problem:

$$\min_{W, \mathbf{a}} \min_{\Omega_r, \Omega_c} \quad \frac{1}{2n} \sum_{i \in [n]} (\hat{y}(\mathbf{x}_i; W, \mathbf{a}) - y_i)^2 + \lambda ||\Omega_r^{1/2} W \Omega_c^{1/2}||_F^2 - \lambda \left( d \log \det(\Omega_r) + p \log \det(\Omega_c) \right)$$

subject to $\quad uI_p \preceq \Omega_r \preceq vI_p, \ uI_d \preceq \Omega_c \preceq vI_d$ (5)

where $\lambda$ is a constant that only depends on $p$ and $d$, $0 < u \leq v$ and $uv = 1$. Note that the constraint is necessary to guarantee the feasible set to be compact so that the optimization problem is well formulated and a minimum is attainable. [2] It is not hard to show that in general the optimization problem (5) is not jointly convex in terms of $\{\mathbf{a}, W, \Omega_r, \Omega_c\}$, and this holds even if the activation function is linear and we do not have the hidden layer. However, as we will show later, for any fixed $\mathbf{a}, W$, the reparametrization makes the partial optimization over $\Omega_r$ and $\Omega_c$ bi-convex. More importantly, we can derive an efficient algorithm that finds the optimal $\Omega_r(\Omega_c)$ for any fixed $\mathbf{a}, W, \Omega_c(\Omega_r)$ in $O(\max\{d^3, p^3\})$ time with closed form solutions. This allows us to apply our algorithm to networks of large sizes, where a typical hidden layer can contain thousands of nodes. Before we delve into the details on solving (5), it is instructive to discuss some of its connections and differences to other learning paradigms.

**Maximum-A-Posteriori Estimation**. Essentially, for model parameter $W$, (5) defines a sequence of MAP problems where each MAP is indexed by the pair of precision matrices $(\Omega_r^{(t)}, \Omega_c^{(t)})$ at iteration $t$. Equivalently, at each stage of the optimization, we can interpret (5) as placing a matrix variate normal prior on $W$ where the precision matrix in the prior is given by $\Omega_r^{(t)} \otimes \Omega_c^{(t)}$. From this perspective, if we fix $\Omega_r^{(t)} = I_p$ and $\Omega_c^{(t)} = I_d, \forall t$, then (5) naturally reduces to learning with $\ell_2$ regularization, or weight decay (Krogh & Hertz, 1992). More generally, for non-diagonal precision matrices, the regularization term for $W$ becomes:

$$||\Omega_r^{1/2} W \Omega_c^{1/2}||_F^2 = ||\text{vec}(\Omega_r^{1/2} W \Omega_c^{1/2})||_2^2 = ||(\Omega_c^{1/2} \otimes \Omega_r^{1/2})\text{vec}(W)||_2^2,$$ (6)

and this is exactly the Tikhonov regularization (Golub et al., 1979) imposed on $W$ where the Tikhonov matrix $\Gamma$ is given by $\Gamma := \Omega_c^{1/2} \otimes \Omega_r^{1/2}$. But instead of manually designing the regularization matrix $\Gamma$ to improve the conditioning of the estimation problem, under the principle of empirical Bayes we propose to also learn both precision matrices (so $\Gamma$ as well) from data.

---

[2]The constraint $uv = 1$ is only for the ease of presentation in the following part and can be readily removed.

**Approximate Volume Minimization**. Let us consider the $\log\det(\cdot)$ function over the positive definite cone. It is well known that the log-determinant function is concave (Boyd & Vandenberghe, 2004). Hence for any pair of matrices $A_1, A_2 \in \mathbb{S}_{++}^m$, the following inequality holds:

$$\log\det(A_1) \leq \log\det(A_2) + \langle \nabla\log\det(A_2), A_1 - A_2 \rangle = \log\det(A_2) + \mathrm{Tr}(A_2^{-1}A_1) - m \quad (7)$$

Applying the above inequality twice by fixing $A_1 = W\Omega_c W^T/2d, A_2 = \Sigma_r$ and $A_1 = W^T\Omega_r W/2p, A_2 = \Sigma_c$ respectively leads to the following inequalities:

$$\begin{cases} d\log\det(W\Omega_c W^T/2d) & \leq -d\log\det(\Omega_r) + \frac{1}{2}\mathrm{Tr}(\Omega_r W\Omega_c W^T) - dp \\ p\log\det(W^T\Omega_r W/2p) & \leq -p\log\det(\Omega_c) + \frac{1}{2}\mathrm{Tr}(\Omega_r W\Omega_c W^T) - dp \end{cases}$$

Using the fact that $\mathrm{Tr}(\Omega_r W\Omega_c W^T) = ||\Omega_r^{1/2}W\Omega_c^{1/2}||_F^2$, we immediately have:

$$d\log\det(W\Omega_c W^T) + p\log\det(W^T\Omega_r W) \leq ||\Omega_r^{1/2}W\Omega_c^{1/2}||_F^2 - (d\log\det(\Omega_r) + p\log\det(\Omega_c)) + c \quad (8)$$

where $c$ is a constant that only depends on $d$ and $p$. Recall that $|\det(A^TA)|$ computes the squared volume of the parallelepiped spanned by the column vectors of $A$. Hence (8) gives us a natural interpretation of the objective function in (5): the regularizer essentially upper bounds the log-volume of the two parallelpipeds spanned by the row and column vectors of $W$. But instead of measuring the volume using standard Euclidean inner product, it also takes into account the local curvatures defined by $\Sigma_r$ and $\Sigma_c$, respectively. For vectors with fixed lengths, the volume of the parallelepiped spanned by them becomes smaller when they are more linearly correlated, either positively or negatively. At a colloquial level, this means that the regularizer in (5) forces fan-in/fan-out of neurons at the same layer to be either positively or negatively correlated with each other, and this corresponds exactly to the effect of *learning from the experience of others*.

### 3.3 THE ALGORITHMS

---

**Algorithm 1** Block Coordinate Descent for Approximate Empirical Bayes

---

**Input:** Initial value $\phi^{(0)} := \{\mathbf{a}^{(0)}, W^{(0)}\}$, $\Omega_r^{(0)} \in \mathbb{S}_{++}^p$ and $\Omega_c^{(0)} \in \mathbb{S}_{++}^d$, first-order optimization algorithm $\mathfrak{A}$, constants $0 < u \leq v$.
 1: **for** $t = 1, \ldots, \infty$ until convergence **do**
 2:     Fix $\Omega_r^{(t-1)}, \Omega_c^{(t-1)}$, optimize $\phi^{(t)}$ by backpropagation and algorithm $\mathfrak{A}$
 3:     $\Omega_r^{(t)} \leftarrow \text{INVTHRESHOLDING}(W^{(t)}\Omega_c^{(t-1)}W^{(t)T}, d, u, v)$
 4:     $\Omega_c^{(t)} \leftarrow \text{INVTHRESHOLDING}(W^{(t)T}\Omega_r^{(t)}W^{(t)}, p, u, v)$
 5: **end for**
 6: ─────────────────────────────────
 7: **procedure** INVTHRESHOLDING$(\Delta, m, u, v)$
 8:     Compute SVD: $Q\mathrm{diag}(\mathbf{r})Q^T = \mathrm{SVD}(\Delta)$
 9:     Hard thresholding $\mathbf{r}' \leftarrow \mathbb{T}_{[u,v]}(m/\mathbf{r})$
10:     **return** $Q\mathrm{diag}(\mathbf{r}')Q^T$
11: **end procedure**

---

In this section we describe a block coordinate descent algorithm to optimize the objective function in (5) and detail how to efficiently solve the matrix optimization subproblems in closed form using tools from convex analysis. Due to space limit, we defer all the proofs to appendix. Given a pair of constants $0 < u \leq v$, we define the following thresholding function $\mathbb{T}_{[u,v]}(x)$:

$$\mathbb{T}_{[u,v]}(x) := \max\{u, \min\{v, x\}\}. \quad (9)$$

We summarize our block coordinate descent algorithm to solve (5) in Alg. 1. In each iteration, Alg. 1 takes a first-order algorithm $\mathfrak{A}$, e.g., the stochastic gradient descent, to optimize the parameters of the neural network by backpropagation. It then proceeds to compute the optimal solutions for $\Omega_r$ and $\Omega_c$ using INVTHRESHOLDING as a sub-procedure. Alg. 1 terminates when a stationary point is found.

We now proceed to show that the procedure INVTHRESHOLDING finds the optimal solution given all the other variables fixed. Due to the symmetry between $\Omega_r$ and $\Omega_c$ in (5), we will only prove this for $\Omega_r$, and similar arguments can be applied to $\Omega_c$ as well. Fix both $W, \Omega_c$ and ignore all the terms that do not depend on $\Omega_r$, the sub-problem on optimizing $\Omega_r$ becomes:

$$\min_{\Omega_r} \quad \mathrm{Tr}(\Omega_r W\Omega_c W^T) - d\log\det(\Omega_r), \qquad \text{subject to} \quad uI_p \preceq \Omega_r \preceq vI_p. \quad (10)$$

**Proposition 1.** The optimization problem (10) is convex.

Define the constraint set $\mathcal{C} := \{A \in \mathbb{S}_{++}^p \mid uI_p \preceq A \preceq vI_p\}$ and the indicator function $\mathbb{I}_{\mathcal{C}}(A) = 0$ iff $A \in \mathcal{C}$ else $\infty$. Given the convexity of (10), we can use the indicator function to first transform (10) into an unconstrained one and use the first-order optimality condition to characterize the optimal solution:

$$0 \in \partial \left( \frac{1}{d} \operatorname{Tr}(\Omega_r W \Omega_c W^T) - \log \det(\Omega_r) + \mathbb{I}_{\mathcal{C}}(\Omega_r) \right) = W \Omega_c W^T / d - \Omega_r^{-1} + \mathcal{N}_{\mathcal{C}}(\Omega_r), \quad (11)$$

where $\mathcal{N}_{\mathcal{C}}(A) := \{B \in \mathbb{S}^p \mid \operatorname{Tr}(B^T(Z - A)) \leq 0, \forall Z \in \mathcal{C}\}$ is the normal cone w.r.t. $\mathcal{C}$ at $A$. Equivalently, we have $\Omega_r^{-1} - W \Omega_c W^T / d \in \mathcal{N}_{\mathcal{C}}(\Omega_r)$. The following key lemma characterizes the structure of the normal cone w.r.t. $\mathcal{C}$:

**Lemma 1.** Let $\Omega_r \in \mathcal{C}$, then $\mathcal{N}_{\mathcal{C}}(\Omega_r) = -\mathcal{N}_{\mathcal{C}}(\Omega_r^{-1})$.

Lemma 1 implies $W \Omega_c W^T / d - \Omega_r^{-1} \in \mathcal{N}_{\mathcal{C}}(\Omega_r^{-1})$. Geometrically, this means that the optimum $\Omega_r^{-1}$ is the Euclidean projection of $W \Omega_c W^T / d$ onto $\mathcal{C}$. Hence it suffices if we can solve the following Euclidean projection problem efficiently, where $\widetilde{\Omega_r} \in \mathbb{S}^p$ is a fixed real symmetric matrix:

$$\min_{\Omega_r} \quad ||\Omega_r - \widetilde{\Omega_r}||_F^2, \qquad \text{subject to} \quad uI_p \preceq \Omega_r \preceq vI_p \tag{12}$$

**Theorem 1.** Let $\widetilde{\Omega_r} \in \mathbb{S}^p$ with eigendecomposition as $\widetilde{\Omega_r} = Q\Lambda Q^T$ and $\operatorname{Proj}_{\mathcal{C}}(\cdot)$ be the Euclidean projection operator onto $\mathcal{C}$, then $\operatorname{Proj}_{\mathcal{C}}(\widetilde{\Omega_r}) = Q\mathbb{T}_{[u,v]}(\Lambda)Q^T$.

**Corollary 1.** Let $W \Omega_c W^T$ be eigendecomposed as $Q\operatorname{diag}(\mathbf{r})Q^T$, then the optimal solution to (10) is given by $Q\mathbb{T}_{[u,v]}(d/\mathbf{r})Q^T$.

Similar arguments can be made to derive the solution for $\Omega_c$ in (5). The final algorithm is very simple as it only contains one SVD, hence its time complexity is $O(\max\{d^3, p^3\})$. Note that the total number of parameters in the network is at least $dp$, hence the algorithm is efficient as it scales sub-quadratically in terms of number of parameters in the network.

## 4 Experiments

So far we develop our model and algorithms based on a simple neural network with one hidden layer and a single output. However, it is straightforward to extend the AEB framework to more sophisticated models with various structures. In this section we demonstrate the effect of our AEB method on learning practical deep neural networks.

### 4.1 Experimental Setup

**Multiclass Classification (MNIST & CIFAR10)**: In the experiments, we show that AEB provides an effective regularization on the network parameters. To this end, we use a convolutional neural network as our baseline model. MNIST considers the following structure: $\text{CONV}_{5\times5\times1\times10}$-$\text{CONV}_{5\times5\times10\times20}$-$\text{FC}_{320\times50}$-$\text{FC}_{50\times10}$. The notation $\text{CONV}_{5\times5\times1\times10}$ denotes the convolutional layer with kernel size $5 \times 5$ from depth 1 to 10; the notation $\text{FC}_{320\times50}$ denotes the fully connected layer with size $320 \times 50$. Similarly, CIFAR10 considers the structure: $\text{CONV}_{5\times5\times3\times10}$-$\text{CONV}_{5\times5\times10\times20}$-$\text{FC}_{500\times500}$-$\text{FC}_{500\times500}$-$\text{FC}_{500\times10}$. To show the effect of regularization, we gradually increase the training set size. MNIST considers the step from 60 to 60,000 (11 different experiments) and CIFAR10 considers the step from 5,000 to 50,000 (10 different experiments). For each training set size, we repeat the experiments for 10 times. The mean along with its standard deviation are shown as the statistics. Moreover, since both the optimization and generalization of neural networks are sensitive to the size of minibatches (Keskar et al., 2016; Goyal et al., 2017), we study two minibatch settings for 256 and 2048, respectively. In our AEB model, we place a matrix-variate normal prior over the weight matrix of the last softmax layer, and we use Alg. 1 to optimize both the model weights of the convolutional network and two covariance matrices of the weight matrix in the last layer.

**Multitask Regression (SARCOS)**: SARCOS relates to an inverse dynamics problem for a seven degree-of-freedom (DOF) SARCOS anthropomorphic robot arm (Vijayakumar & Schaal, 2000). The

goal of this task is to map from a 21-dimensional input space (7 joint positions, 7 joint velocities, 7 joint accelerations) to the corresponding 7 joint torques. Hence there are 7 tasks and the inputs are shared among all the tasks. The training set and test set contain 44,484 and 4,449 examples, respectively. The network structure is given by $FC_{21 \times 256}$-$FC_{256 \times 100}$-$FC_{100 \times 7}$. Again, we apply our AEB method on the last layer weight matrix, where each row corresponds to a separate task vector.

We compare our AEB method with classic regularization methods in the literature, including weight decay, dropout (Srivastava et al., 2014), batch normalization (BN) (Ioffe & Szegedy, 2015) and the DeCov method (Cogswell et al., 2015). We also note that we fix all the hyperparameters such as learning rate to be the same for all the methods. To better understand the working mechanism of the proposed method, we report evaluation metrics on test set as a measure of generalization, the trajectory of the loss function during training, and the correlation of the weight matrix.

### 4.2 RESULTS

**Multiclass Classification (MNIST & CIFAR10)**: Results on the multiclass classification for different training sizes are show in Fig. 2. For both MNIST and CIFAR10, we find AEB, Weight Decay, and Dropout are the effective regularization methods, while Batch Normalization and DeCov vary in different settings. Batch Normalization suffers from large batch size in CIFAR10 (comparing Fig. 2 (c) and (d)) but is not sensitive to batch size in MNIST (comparing Fig. 2 (a) and (b)). The performance deterioration in large batch size of Batch Normalization is also observed by Hoffer et al. (2017). DeCov, on the other hand, improves the generalization in MNIST with batch size 256 (see Fig. 2 (a)), while it demonstrates only comparable or even worse performance in other settings. To conclude, as training set size grows, AEB consistently performs better generalization as comparing to other regularization methods. We also note that AEB is not sensitive to the size of minibatches while most of the methods suffer from large minibatches. In appendix, we show the combination of AEB with other generalization methods can usually lead to even better results.

**Multitask Regression (SARCOS)**: In this experiment we are interested in investigating whether our AEB method can lead to better generalization for multiple related regression problems. To do so, we report the explained variance as a normalized metric, e.g., one minus the ratio between mean squared error and the variance of different methods in Table 1. The larger the explained variance, the better the predictive performance. In this case we observe a consistent improvement of AEB over other competitors on all the 7 regression tasks. We would like to emphasize that all the experiments share exactly the same experimental protocol, including network structure, optimization algorithm, training iteration, etc, so that the performance differences can only be explained by different ways of regularizations. For better visualization, we also plot the result in appendix.

**Optimization**: It has recently been empirically shown that BN helps optimization not by reducing internal covariate shift, but instead by smoothing the landscape of the loss function (Santurkar et al., 2018). To understand how AEB improves generalization, in Fig. 3, we plot the values of the cross entropy loss function on both the training and test sets during optimization using Alg. 1. The experiment is performed in MNIST with batch size 2048. In this experiment, we fix the number of outer loop to be 2 and each block optimization over network weights contains 50 epochs. Because of the stochastic optimization over model weights, we can see several unstable peaks in function value around iteration 50 when trained with AEB, which corresponds to the transition phase between two consecutive outer loops with different row/column covariance matrices. In both cases AEB converges to better local optima of the loss landscape, which lead to better generalization on the test set as well because they have smaller loss values on the test set when compared with training without AEB.

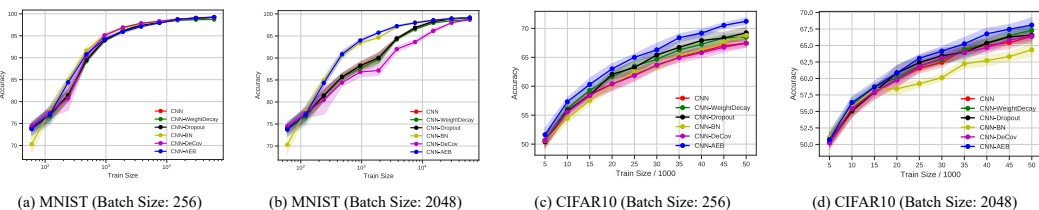

|                        |                        |                           |                           |
| :--------------------: | :--------------------: | :-----------------------: | :-----------------------: |
| (a) MNIST (Batch Size: 256) | (b) MNIST (Batch Size: 2048) | (c) CIFAR10 (Batch Size: 256) | (d) CIFAR10 (Batch Size: 2048) |

Figure 2: Generalization of AEB on MNIST and CIFAR10. AEB improves generalization under both minibatch settings and is most beneficial when training set is small.

Table 1: Explained variance of different methods on 7 regression tasks from the SARCOS dataset.

| Method | 1st | 2nd | 3rd | 4th | 5th | 6th | 7th |
|---|---|---|---|---|---|---|---|
| **MTL** | 0.4418 | 0.3472 | 0.5222 | 0.5036 | 0.6024 | 0.4727 | 0.5298 |
| **MTL-Dropout** | 0.4413 | 0.3271 | 0.5202 | 0.5063 | 0.6036 | 0.4711 | 0.5345 |
| **MTL-BN** | 0.4768 | 0.3770 | 0.5396 | 0.5216 | 0.6117 | 0.4936 | 0.5479 |
| **MTL-DeCoV** | 0.4027 | 0.3137 | 0.4703 | 0.4515 | 0.5229 | 0.4224 | 0.4716 |
| **MTL-AEB** | **0.4769** | **0.3969** | **0.5485** | **0.5308** | **0.6202** | **0.5085** | **0.5561** |

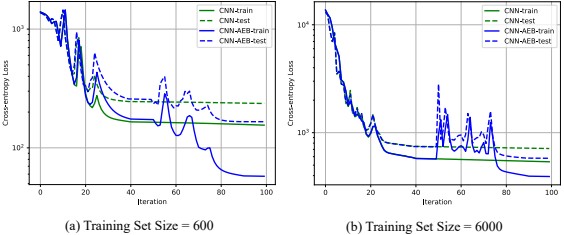

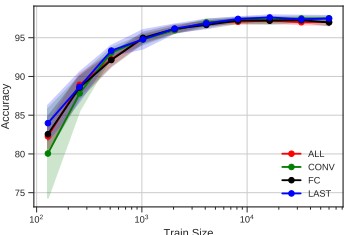

Figure 3: Optimization of AEB on MNIST with batch size 2048. Sequential Tikhonov regularizations in AEB help to converge to better local optima during training.

Figure 4: Applying AEB on different layers in neural networks for MNIST with batch size 2048.

**Stable rank and spectral norm**: Given a matrix $W$, the stable rank of $W$, denoted as s-rank$(W)$, is defined as s-rank$(W) := ||W||_F^2/||W||_2^2$. In other words, the stable rank of a matrix is the ratio of its squared Frobenius norm and squared spectral norm. Clearly, for any matrix $W$, we have $1 \leq$ s-rank$(W) \leq$ rank$(W)$. As its name suggests, the stable rank is more stable than the rank because it is largely unaffected by tiny singular values. Let $r =$ rank$(W)$, then s-rank$(W) =$ rank$(W)$ when $\sigma_1(W) = \cdots = \sigma_r(W)$ and s-rank$(W) = 1$ when rank$(W) = 1$. It has recently been shown (Neyshabur et al., 2017, Theorem 1) that the generalization error of neural networks crucially depends on both the stable ranks and the spectral norms of connection matrices in the network. Specifically, it can be shown that the generalization error is upper bounded by $O\big(\sqrt{\prod_{j=1}^{L}||W_j||_2^2 \sum_{j=1}^{L} \text{s-rank}(W_j)/n}\big)$, where we omit constant terms that do not depend on weight matrices and $L$ is the number of layers in the network. Essentially, this upper bound suggests that smaller spectral norm (smoother function mapping) and stable rank (skewed spectrum) leads to better generalization.

To understand why AEB improves generalization, in Fig. 5 and Fig. 6, we plot both the stable rank and the spectral norm of the weight matrix in the last layer of the CNNs used in our MNIST and CIFAR10 experiments. We compare 3 methods: CNN without any regularization, CNN trained with weight decay and CNN with AEB. For each setting we repeat the experiments for 5 times, and we plot the mean along with its standard deviation. From Fig. 5a and Fig. 6a it is clear that AEB leads to a significant reduction in terms of the stable rank when compared with weight decay, and this effect is consistent in all the experiments with different training size. Similarly, in Fig. 5b and Fig. 6b we plot the spectral norm of the weight matrix. Again, both weight decay and AEB help reduce the spectral norm in all settings, but AEB plays a more significant role than the usual weight decay. Combining the experiments with the generalization upper bound introduced above, we can see that training with AEB leads to an estimator of $W$ that has lower stable rank and smaller spectral norm, which explains why it achieves a better generalization performance. Furthermore, this observation holds on the SARCOS datasets as well, and we show the results in the appendix.

**Ablations**: In all the experiments, the AEB algorithm is performed on the softmax layer. Here, we study the effects of applying AEB algorithm in all CONV/FC layers, all CONV layers, all FC layers, and the last FC layer (i.e., softmax layer). We first discuss how we handle the convolutions in our AEB algorithm. Consider a convolutional layer with {input channel, output channel, kernel width, kernel height} being $\{a, b, k_w, k_h\}$, we vectorize the original 4-D tensor to be a 2-D matrix of size $ak_wk_h \times b$. The AEB algorithm can therefore be directly applied on this transformed matrix. Next, we perform the experiment on MNIST with batch size 2048 in Fig. 4. The training set size here is chosen as {128, 256, 512, 1024, 2048, 4096, 8192, 16384, 32768, 60000}.

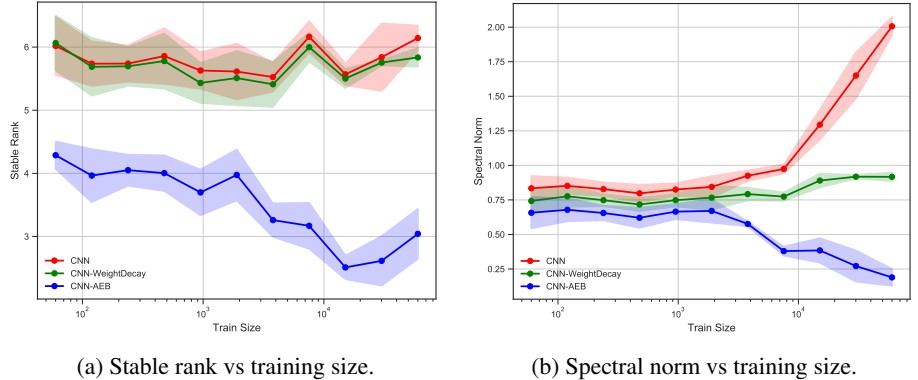

(a) Stable rank vs training size.          (b) Spectral norm vs training size.

Figure 5: Comparisons of stable ranks and spectral norms from different methods on MNIST.

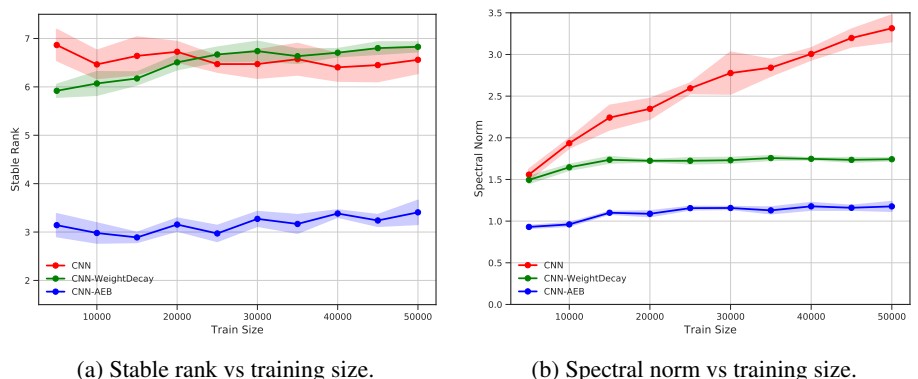

(a) Stable rank vs training size.          (b) Spectral norm vs training size.

Figure 6: Comparisons of stable ranks and spectral norms from different methods on CIFAR10.

We find that simply applying the AEB algorithm in the softmax layer reaches best generalization as comparing to applying AEB on more layers. The improvement is more obvious when the training set size is small. We argue that neural networks can be realized as a combination of a complex nonlinear transformation (i.e., feature extraction) and a linear model (i.e., softmax layer). Since AEB represents a correlation learning in the weight matrix, it implies that implicit correlations of neurons can also be discovered. In the real world setting, different tasks should be correlated. Therefore, applying AEB in the linear model shall improve the model performance by discovering these tasks correlations. On the contrary, the nonlinear features should be decorrelated for the purpose of generalization (Cogswell et al., 2015). Hence, applying AEB in previous layers may lead to adversarial effect.

**Correlation Matrix**: To verify that AEB imposes the effect of "learning from the experience of others" during training, we visualize the weight matrix of the softmax layer by computing the corresponding correlation matrix, as shown in Fig. 7. In Fig. 7, darker color means stronger correlation. We conduct two experiments with training size 600 and 60,000 respectively. As we can observe, training with AEB leads to weight matrix with stronger correlations, and this effect is more evident when the training set is large. This is consistent with our analysis that AEB encourages learning from the experience of others. As a sanity check, from Fig. 7 we can also see that similar digits, e.g., 1 and 7, share a positive correlation while dissimilar ones, e.g., 1 and 8, share a negative correlation.

# 5 RELATED WORK

Different kinds of regularization approaches have been studied and designed for neural networks, e.g., weight decay (Krogh & Hertz, 1992), early stopping (Caruana et al., 2001), Dropout (Srivastava et al., 2014) and the more recent DeCov (Cogswell et al., 2015) method. BN was proposed to reduce the internal covariate shift during training, but recently it has been empirically shown to actually smooth the landscape of the loss function (Santurkar et al., 2018). As a comparison, we propose

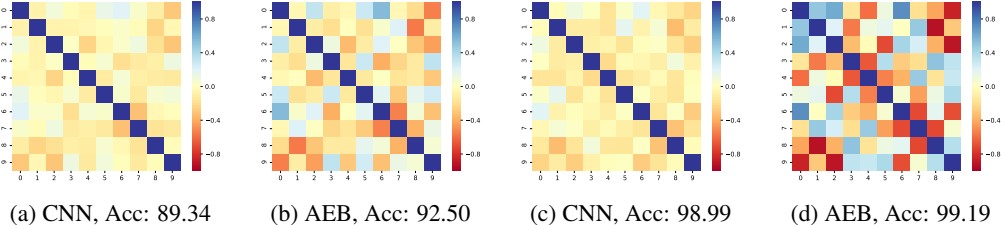

(a) CNN, Acc: 89.34  (b) AEB, Acc: 92.50  (c) CNN, Acc: 98.99  (d) AEB, Acc: 99.19

Figure 7: Correlation matrix of the weight matrix in the softmax layer. All the networks share the same structure and optimization protocol. The left two correspond to dataset with training size 600 and the right two with size 60,000. Acc means the test set classification accuracy.

AEB through the lens of the empirical Bayes method, with the aim to reduce overfitting by allowing neurons to learn from each other. From the optimization perspective, learning the row and column covariance matrices help to converge to better local optimum that also generalizes better.

Despite the name, empirical Bayes method is in fact a frequentist approach to obtain estimator with favorable properties. On the other hand, truly Bayesian inference would instead put a (approximate/variational) posterior distribution over model weights to characterize the uncertainty during training (MacKay, 1992; Hernández-Lobato & Adams, 2015; Blundell et al., 2015). However, due to the complexity of nonlinear neural networks, analytic posterior is not available, hence strong independent assumptions over model weight have to be made in order to achieve computationally tractable variational solution. Typically, both the prior and the variational posterior are assumed to fully factorize over model weights. As a comparison, in AEB we characterize the correlation between model weights via learned covariance matrix with a Kronecker product structure, which is more flexible and realistic. The application of empirical Bayes in the machine learning community is not new. McInerney (2017) proposed Bayes Empirical Bayes method for choosing the hyper-hyperparameters of models, a.k.a., the Type-III MLE method. It proposes to use Monte Carlo sampling to approximate both the marginal likelihood and the prior distribution, in order to optimize the hyper-prior distribution over the hyper-hyperparameters. As a comparison, in our work we study the matrix-variate prior with Kronecker factorization structure and we propose an analytic solution that could be computed in closed form without any sampling.

Orthogonal to our work, the Kronecker factorization assumption has also been applied in the literature of neural networks to approximate the Hessian matrix, or the Fisher information matrix, in second-order optimization methods (Martens & Grosse, 2015; Zhang et al., 2017). The main idea here is to approximate the curvature of the loss function's landscape, in order to achieve better convergence speed compared with first-order method while maintaining the tractability of such computation.

Determinantal point process (DPP) has been previously applied to compress neural networks (Mariet & Sra, 2015). Specifically, a DPP kernel is placed over the activations of neurons from the same layer, and then neurons with similar activations over a fixed dataset are merged into a single one. However, it is well known that DPPs can capture only negative correlations (Kulesza & Taskar, 2011; Kulesza et al., 2012), and as a result they do not stimulate neurons to learn from the experience of other neurons. As a comparison, by explicitly learning both precision (covariance) matrices, our framework can account for both positive and negative correlations among fan-in/fan-out of neurons from the same layer.

## 6 CONCLUSION

In this paper we propose an approximate empirical Bayes method with matrix-variate normal prior to learn the model parameters of deep neural networks. The prior encourages neurons to borrow statistical strength from other neurons during the learning process, and it provides an effective regularization when training large networks on small datasets. We connect our approach with sequential MAP inference and volume minimization, and we also design an efficient block coordinate descent algorithm to optimize the model. Empirically, on three datasets we demonstrate that our AEB method improves generalization by finding better local optima that does not overfit. One future direction is to develop a better approximate solution to optimize the two covariance matrices from the marginal log-likelihood function.

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

## A    PROOFS

**Proposition 1.** The optimization problem (10) is convex.

*Proof.* It is clear that the objective function is convex: the trace term is linear in $\Omega_r$ and it is well-known that the $\log \det(\cdot)$ is concave in the positive definite cone (Boyd & Vandenberghe, 2004), hence it trivially follows that $\mathrm{Tr}(\Omega_r W \Omega_c W^T) - d \log \det(\Omega_r)$ is convex in $\Omega_r$.

It remains to show that the constraint set is also convex. Let $\Omega_1, \Omega_2$ be any feasible points, i.e., $uI_p \preceq \Omega_1 \preceq vI_p$ and $uI_p \preceq \Omega_2 \preceq vI_p$. Let $\forall t \in (0, 1)$, we have:

$$||t\Omega_1 + (1-t)\Omega_2||_2 \le t||\Omega_1||_2 + (1-t)||\Omega_2||_2 \le tv + (1-t)v = v,$$

where we use $|| \cdot ||_2$ to denote the spectral norm of a matrix. Now since both $\Omega_1$ and $\Omega_2$ are positive definite, the spectral norm is also the largest eigenvalue, hence this shows that $t\Omega_1 + (1-t)\Omega_2 \preceq vI_p$.

To show the other direction, we use the Courant-Fischer characterization of eigenvalues. Let $\lambda_{\min}(A)$ denote the minimum eigenvalue of a real symmetric matrix $A$, then by the Courant-Fischer min-max theorem, we have:

$$\lambda_{\min}(A) := \min_{\mathbf{x} \ne 0, ||\mathbf{x}||_2 = 1} ||A\mathbf{x}||_2.$$

For the matrix $t\Omega_1 + (1-t)\Omega_2$, let $\mathbf{x}^*$ be the vector corresponding to the minimum eigenvalue, hence we have:

$$\begin{aligned}
\lambda_{\min}(t\Omega_1 + (1-t)\Omega_2) &= \min_{\mathbf{x} \ne 0, ||\mathbf{x}||_2 = 1} ||(t\Omega_1 + (1-t)\Omega_2)\mathbf{x}||_2 \\
&= (t\Omega_1 + (1-t)\Omega_2)\mathbf{x}^* \\
&\ge t\lambda_{\min}(\Omega_1) + (1-t)\lambda_{\min}(\Omega_2) \\
&\ge tu + (1-t)u \\
&= u,
\end{aligned}$$

which also means that $t\Omega_1 + (1-t)\Omega_2 \succeq uI_p$, and this completes the proof. ∎

**Lemma 1.** Let $\Omega_r \in \mathcal{C}$, then $\mathcal{N}_{\mathcal{C}}(\Omega_r) = -\mathcal{N}_{\mathcal{C}}(\Omega_r^{-1})$.

*Proof.* Let $S \in \mathcal{N}_{\mathcal{C}}(\Omega_r)$. We want to show $-S \in \mathcal{N}_{\mathcal{C}}(\Omega_r^{-1})$. By definition of the normal cone, since $S \in \mathcal{N}_{\mathcal{C}}(\Omega_r)$, we have:

$$\mathrm{Tr}(SZ) \le \mathrm{Tr}(S\Omega_r), \quad \forall Z \in \mathcal{C}$$

Now realize that $\Omega_r \in \mathcal{C}$ and $\mathcal{C}$ is a compact set, it follows $\Omega_r$ is the solution of the following linear program:

$$\max \quad \mathrm{Tr}(SZ), \qquad \text{subject to} \quad Z \in \mathcal{C}$$

Since both $S$ and $Z$ are real symmetric matrix, we can decompose them as $Z := Q_Z \Lambda_Z Q_Z^T$ and $S := Q_S \Lambda_S Q_S^T$, where both $Q_Z, Q_S$ are orthogonal matrices and $\Lambda_Z, \Lambda_S$ are diagonal matrices with the corresponding eigenvalues in decreasing order. Plug them into the objective function, we have:

$$\mathrm{Tr}(SZ) = \mathrm{Tr}(Q_S \Lambda_S Q_S^T Q_Z \Lambda_Z Q_Z^T) = \mathrm{Tr}(\Lambda_S Q_S^T Q_Z \Lambda_Z Q_Z^T Q_S).$$

Define $K := Q_S^T Q_Z$ and $D = K \circ K$, where we use $\circ$ to denote the Hadamard product between two matrices. Since both $Q_S$ and $Q_Z$ are orthogonal matrices, we know that $K$ is also orthogonal, which implies:

$$\sum_{j=1}^{p} D_{ij} = 1, \forall i \in [p], \quad \text{and} \quad \sum_{i=1}^{p} D_{ij} = 1, \forall j \in [p].$$

As a result, $D$ is a doubly stochastic matrix and we can further simplify the objective function as:

$$\mathrm{Tr}(\Lambda_S Q_S^T Q_Z \Lambda_Z Q_Z^T Q_S) = \mathrm{Tr}(\Lambda_S K \Lambda_Z K^T) = \lambda_S^T D \lambda_Z = \sum_{i,j=1}^{p} \lambda_{S,i} D_{ij} \lambda_{Z,j},$$

where $\lambda_S$ and $\lambda_Z$ are $p$ dimensional vectors that contain the eigenvalues of $S$ and $Z$ in decreasing order, respectively. Now for any $\lambda_S$ and $\lambda_Z$ in decreasing order, we have:

$$u \sum_{i=1}^{p} \lambda_{S,i} \leq \sum_{i=1}^{p} \lambda_{S,i} \lambda_{Z,1+p-i} \leq \sum_{i,j=1}^{p} \lambda_{S,i} D_{ij} \lambda_{Z,j} \leq \sum_{i=1}^{p} \lambda_{S,i} \lambda_{Z,i} \leq v \sum_{i=1}^{p} \lambda_{S,i} \qquad (13)$$

From (13), in order for $\Omega_r$ to maximize the linear program, it must hold that $D = K = I_p$ and all the eigenvalues of $\Omega_r$ are $v$. But due to the assumption that $uv = 1$, in this case we also know that all the eigenvalues of $\Omega_r^{-1}$ are $1/v = u$, hence $\Omega_r^{-1}$ also minimizes the above linear program, which implies:

$$\text{Tr}(S\Omega_r^{-1}) \leq \text{Tr}(SZ), \quad \forall Z \in \mathcal{C} \Leftrightarrow \text{Tr}(-S(Z - \Omega_r^{-1})) \leq 0 \quad \forall Z \in \mathcal{C}.$$

In other words, we have $-S \in \mathcal{N}_\mathcal{C}(\Omega_r^{-1})$. Using exactly the same arguments it is clear to see that the other direction also holds, hence we have $\mathcal{N}_\mathcal{C}(\Omega_r) = -\mathcal{N}_\mathcal{C}(\Omega_r^{-1})$. ∎

**Theorem 1.** Let $\widetilde{\Omega_r} \in \mathbb{S}^p$ with eigendecomposition as $\widetilde{\Omega_r} = Q\Lambda Q^T$ and $\text{Proj}_\mathcal{C}(\cdot)$ be the Euclidean projection operator onto $\mathcal{C}$, then $\text{Proj}_\mathcal{C}(\widetilde{\Omega_r}) = Q\mathbb{T}_{[u,v]}(\Lambda)Q^T$.

*Proof.* Since $\Omega_r \in \mathcal{C}$ is real and symmetric, we can reparametrize $\Omega_r$ as $\Omega_r := U\Lambda_{\Omega_r}U^T$ where $U$ is an orthogonal matrix and $\Lambda_{\Omega_r}$ is a diagonal matrix whose entries corresponds to the eigenvalues of $\Omega_r$. Recall that $U$ corresponds to a rigid transformation that preserves length, so we have:

$$||\Omega_r - \widetilde{\Omega_r}||_F^2 = ||U\Lambda_{\Omega_r}U^T - UU^T\widetilde{\Omega_r}UU^T||_F^2 = ||\Lambda_{\Omega_r} - U^T\widetilde{\Omega_r}U||_F^2 \qquad (14)$$

Define $B := U^T\widetilde{\Omega_r}U$. Now by using the fact that $\widetilde{\Omega_r}$ can be eigendecomposed as $\widetilde{\Omega_r} = Q\Lambda Q^T$, we can further simplify (14) as:

$$||\Lambda_{\Omega_r} - U^T\widetilde{\Omega_r}U||_F^2 = \sum_{i \in [p]} (\Lambda_{\Omega_r,ii} - B_{ii})^2 + \sum_{i \neq j} B_{ij}^2 \geq \sum_{i \in [p]} (\Lambda_{\Omega_r,ii} - B_{ii})^2 \geq \sum_{i \in [p]} (\mathbb{T}_{[u,v]}(B_{ii}) - B_{ii})^2,$$

where the last inequality holds because $u \leq \Lambda_{\Omega_r,ii} \leq v, \forall i \in [p]$. In order to achieve the first equality, $B = U^T\widetilde{\Omega_r}U$ should be a diagonal matrix, which means $U^TQ = I_p \Leftrightarrow U = Q$. In this case, $\text{diag}(B) = \Lambda$. To achieve the second equality, simply let $\Lambda_{\Omega_r} = \mathbb{T}_{[u,v]}(\text{diag}(B)) = \mathbb{T}_{[u,v]}(\Lambda)$, which completes the proof. ∎

## B  MORE EXPERIMENTS

In this section we present more experimental results.

**Combination**. As discussed in the main text, combining the proposed AEB with BN can further improve the generalization performance, due to the complementary effects between these two approaches: BN helps smoothing the landscape of the loss function while AEB also changes the curvature via the row and column covariance matrices (see Fig. 8).

On the other hand, we do not observe significant difference when combining AEB with Dropout on this dataset. While we are not clear what is the exact reason for this effect, we conjecture this is due to the fact that Dropout works as a regularizer that prevents coadaptation while AEB instead encourages neurons to learn from each other.

**Covariance matrices in the prior**. One byproduct that AEB brings to us is the learned row and column covariance matrices, which can be used in exploratory data analysis to understand the correlations between learned features and different output tasks. To this end, we visualize both the row and column covariance matrices in Fig. 9. The two covariance matrices on the first row correspond to the ones learned on a training set with 600 instances while the two on the second row are trained with the full dataset on MNIST.

From Fig. 9 we can make the following observations: the structure of both covariance matrices become more evident when trained with larger dataset, and this is consistent with the Bayesian principle because more data provide more evidence. Second, we observe in our experiments that the variances of both matrices are small. In fact, the variance of the row covariance matrix $\Sigma_r$ achieves the lower bound limit $u$ at convergence, which also validates our approximation in computing the

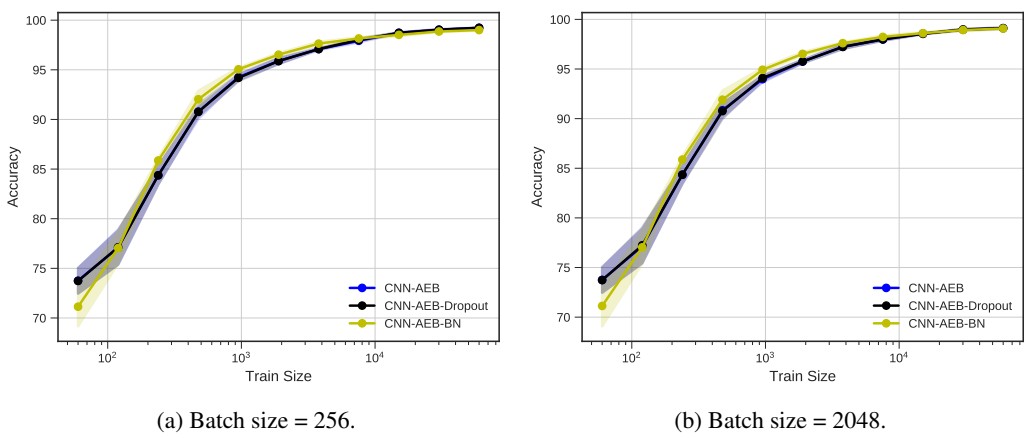

(a) Batch size = 256.

(b) Batch size = 2048.

Figure 8: Combine AEB with BN and Dropout on MNIST.

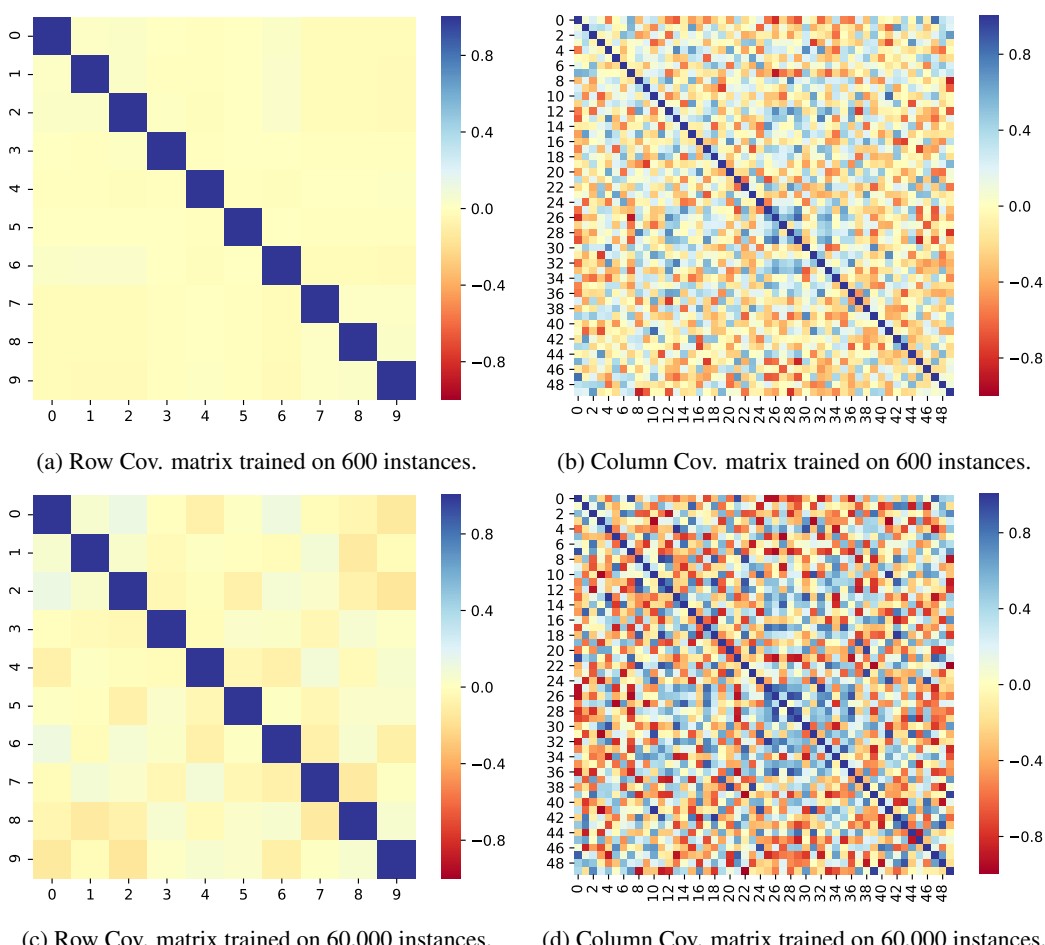

(a) Row Cov. matrix trained on 600 instances.

(b) Column Cov. matrix trained on 600 instances.

(c) Row Cov. matrix trained on 60,000 instances.

(d) Column Cov. matrix trained on 60,000 instances.

Figure 9: Recovered row covariance matrix $\Sigma_r$ and column covariance matrix $\Sigma_c$ in the prior distribution on MNIST.

marginal distribution in (2). Lastly, comparing the row covariance matrix $\Sigma_r$ in Fig. 9 with the one computed from model weights in Fig. 7, we can see that both matrices exhibit the same correlation patterns, except that the one obtained from model weights are more evident, which is due to the

fact that model weights are closer to data evidence than the row covariance matrix in our Bayesian hierarchy.

On the other hand, the column covariance matrix in Fig. 9 also exhibit rich correlations between the learned features, e.g., the neurons in the penultimate layer. Again, with more data, these patterns become more evident.

**Stable rank and spectral norm**. In this paragraph we show the experimental results of stable ranks and spectral norms on the SARCOS dataset. For the SARCOS dataset, the weight matrix being regularized is of dimension $100 \times 7$. Again, we compare the results using three methods: MTL, MTL-WeightDecay and MTL-AEB. All the network architectures used in this experiment are consistent with the ones used before.

Table 2: Stable rank and spectral norm on the SARCOS dataset.

|                  | Stable Rank | Spectral Norm |
| ---------------- | ----------- | ------------- |
| MTL              | 4.48        | 0.96          |
| MTL-WeightDecay  | 4.83        | 0.92          |
| MTL-AEB          | **2.88**    | **0.70**      |

As can be observed from Table 2, compared with the weight decay regularization, our AEB regularization greatly reduces both the stable rank and the spectral norm of learned weight matrix, which also helps to explain why MTL-AEB generalizes better compared with MTL and MTL-WeightDecay.

To conclude this section, we plot a bar chart in Fig. 10 to show the explained variance achieved by different methods on 7 regression tasks from the SARCOS dataset for better visualization. Again, we can see that AEB improves uniformly over all the other methods on all the 7 related regression tasks.

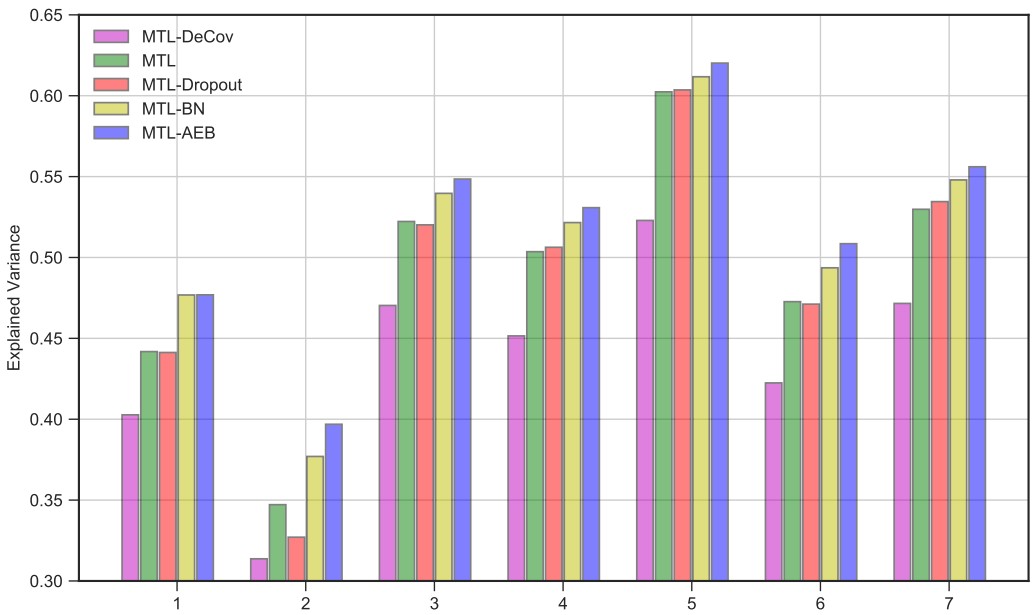

Figure 10: Explained variance of different methods on 7 regression tasks from the SARCOS dataset.

