# OpenReview forum: "Learning From the Experience of Others: Approximate Empirical Bayes in Neural Networks"
_ICLR.cc/2019/Conference_

### Official Review · AnonReviewer2 · 2018-11-02
**The paper presents a novel approach to addressing the problem of overfitting when training large networks on small data sets. The main idea is to use a dependent prior on the network weights (instead of commonly used independent normal priors) in order to allow the weights "learning from the experience of others".**

**Rating:** 7
**Confidence:** 4

**Review:**

Starting from a simple neural network with only one hidden layer and a single output, the basic idea of approximate empirical Bayes (AEB) method is proposed, defining a matrix-variate normal prior distribution with a Kronecker product structure, so as to capture correlations between the row and column vectors of the weight matrix. Then, a block coordinate descent algorithm for solving the optimization problem is proposed. It consists of alternating three steps to obtain the optimal solutions of model parameters, row and column covariance matrices.

The current method is investigated and tested on three data sets for both classification (MNIST & CIFAR10) and regression (SARCOS) tasks. Encouraging experimental results demonstrate that the correlation learning in the weight matrix significantly improves performance when the training set size is relatively small. It is also shown that the proposed AEB method does not seem sensitive to the size of mini-batches and its combination with other generalization methods can lead to better results in some cases.

 Strengths:

 This paper is mostly well written and overall is easy to follow. It clearly reveals that correlation in the weight matrix plays a crucial role in better generalizing on small training sets.

 Minor comments:

 * The authors state that it is straightforward to extend the proposed method to more sophisticated models with various structures, such as CNN. Perhaps a bit more detail should be given in the main text.
 * Fig. 6 on page 12 is not explicitly mentioned in the main text. It seems a bit confusing.

---

> ### Author Response · Authors · 2018-11-08
> **Thanks for the encouraging comments**
>
> We would like to thank the reviewer for the encouraging comments. The extension of the proposed method to the convolution layers in CNNs is described in the first paragraph of the ablation study in the experiment section. Fig. 6 is used to illustrate the effects of combining the proposed method with BN and Dropout. Thanks for pointing out this, and we will update the paper accordingly.

---

### Official Review · AnonReviewer1 · 2018-11-08
**Kronecker-factored weight-decay with parameters learned via empirical Bayes has the potential to be better than standard L2 weight decay, but the novelty, motivation and empirical evaluation are not convincing**

**Rating:** 3
**Confidence:** 5

**Review:**

Summary: The submission proposes a method to learn Kronecker factors of the covariance of a matrix-variate normal distribution over neural network weights. Given a setting of the neural network weights, the Kronecker factors can be found in closed form by solving a convex optimization problem.

Strengths:
+ The paper provides a thorough theoretical motivation for computing a positive definite factorization of the regularization term in the empirical Bayes setup.

Weaknesses:
- The novelty of the method is overstated.
- The method is claimed to be efficient, but each iteration requires an inner loop of solving a MAP problem (via gradient descent on the negative log likelihood with an extra regularization term), which is at least as expensive as a standard training run.
- The submission lacks precise technical writing, and many technical details appear in inappropriate places, such as the introduction.
- The experimental evaluation is not strong.

Major comments:
- The use of empirical Bayes is not novel in the context of neural networks despite the submission's claim that "Existing studies on
parametric empirical Bayes methods focus on the setting where the likelihood function and the prior are assumed to have specific forms" (pg. 1). In particular, see e.g., https://papers.nips.cc/paper/6864-an-empirical-bayes-approach-to-optimizing-machine-learning-algorithms.pdf, https://arxiv.org/abs/1801.08930, https://arxiv.org/abs/1807.01613 for the use of non-conjugate likelihoods in empirical Bayes.
- The cited motivation for the use of a matrix-variate normal prior over the weights of a neural network is weak. In particular, one iteration of credit assignment via backpropagation in a one-layer neural network does not adequately describe the complex interactions between parameters of a nonlinear model over the course of an optimization procedure such as the one used in line 2 of Algorithm 1. In addition, a learned prior can be used to introduce additional correlations between parameters, so it is strange to describe the learned prior as "capturing correlations" resulting from a single weight update.
- The submission lacks clarity on the assumptions entailed by using the proposed methodology, namely that the Kronecker factors are assumed positive definite. For instance, the logdet function is defined for positive definite (PD) matrices, the results in the paragraph titled "Approximate Volume Minimization" hold only for PD matrices, and the InvThresholding procedure is valid for only the same class. It should additionally be noted that this assumption is *not* required for a Kronecker factorization to be defined.
- The submission makes heavy use of Kronecker factorization, but neglects to cite works that use a similar factorization of the covariance matrices for neural network applications (e.g., https://arxiv.org/abs/1503.05671, https://arxiv.org/abs/1712.02390). Furthermore, the method bears a strong similarity to https://arxiv.org/abs/1506.02117 in learning a Kronecker-factored covariance structure between parameters of a neural network. Can the authors comment on the similarities and differences?
- Results are reported on a simple regression task (SARCOS) and multiclass classification problems (MNIST & CIFAR10) using a neural network with a single hidden layer. Moreover, "in all the experiments, the AEB algorithm is performed on the softmax layer" (pg. 7) and the justification for this in the "Ablations" section is opaque to me. Was a similar restriction used for L2 weight decay regularization? I can't interpret how thorough the evaluation is without such details. It is also not clear that the approach is extensible to more complex architectures, or that there would be a significant empirical benefit if this is done.
- Figure 5 does not really exhibit interesting learned structure in the correlation matrix. Why not plot a visualization of the learned prior, rather than the weights?

Minor comments:
- The submission needs to be checked for English grammar and style.
- abstract: "Learning deep neural networks could be understood as the combination of representation learning and learning halfspaces." This is unclear.
- pg. 2: "Empirically, we show that the proposed method helps the network
converge to better local optima that also generalize better..."  What is a "better" local optimum?
- Section 3.1 describes empirical Bayes with point estimates. Please make it clearer that this methodology is not itself a contribution of paper by citing prior work.
- pg. 5: "Alg. 1 terminates when a stationary point is found." What exactly is the stopping criterion?
- pg. 6: The labels in Figure 2 are extremely small. Moreover, please keep the y-axis range constant.
- pg. 6, Figure 2 caption: "AEB improves generalization under both minibatch settings and is most beneficial when training set is small." Do the CIFAR10 results not show the opposite effect, that the regularization is most beneficial when the training set is large?
- pg. 7: "Batch Normalization suffers from large batch size in CIFAR10" weird wording
- pg. 8: "One future direction is to develop a better approximate solution to optimize the two covariance matrices from the marginal log-likelihood function." This is unclear.

---

> ### Author Response · Authors · 2018-11-08
> **Some clarifications - Major comments - PartI**
>
> We would like to thank the reviewer for the detailed comments. We hope to take this chance to clarify several misconceptions that the reviewer had about the paper. At a high level, we want to emphasize that the contributions in this paper, including the empirical Bayes framework with Kronecker factorized prior for learning neural nets and the proposed optimization algorithm, are novel.
>
> Major comments:
> - "The use of empirical Bayes is not novel in the context of neural networks..."
> We will highlight the connections and differences of these three works with ours here.
> 1.  "https://papers.nips.cc/paper/6864-an-empirical-bayes-approach-to-optimizing-machine-learning-algorithms.pdf". This paper mainly discusses Bayes Empirical Bayes method for choosing the hyper-hyperparameters of ML models, a.k.a., the Type-III MLE method, see Eq. (3) in the paper. As a comparison, our approach could be understood as a Type-II MLE method. On the technical side, this paper does not deal with matrix-variate prior with Kronecker structure, and it proposes to use Monte Carlo sampling to approximate both the marginal likelihood and the prior distribution (in order to optimize the hyper-prior distribution over \lambda in the paper). While in our work we study the specific matrix-variate prior with Kronecker product that is well motivated from learning neural networks, and we propose an analytic solution that could be computed in closed form to find the MAP, which means we don't use any Monte Carlo samplings.
>
> 2.  "Recasting Gradient-Based Meta-Learning as Hierarchical Bayes". Note that we have already discussed this paper in the last paragraph of page 2. A high-level summary: instead of using point estimate, this paper discusses about approximating the marginal distribution using the Laplacian method. Again, this paper does not talk about the specific Kronecker factorization prior, and the context of this paper is to interpret the meta-learning algorithm from the perspective of EB, which is very interesting, but is orthogonal to our approach.
>
> 3.  "Conditional Neural Processes". To the best of our knowledge, this paper discusses about how to incorporate inference in the Gaussian process into a NN training regime. It does not talk about Empirical Bayes nor Kronecker-product prior.
>
>
> - "In particular, one iteration of credit assignment via backpropagation in a one-layer neural network does not adequately describe the complex interactions between parameters of a nonlinear model over the course of an optimization procedure such as the one used in line 2 of Algorithm 1"
> We would like to point out that Line 2 in Alg. 1 does not only optimize the network in one iteration, instead, it optimizes the network until a stationary point has been found. This can be found in the optimization paragraph in the experiment section and can also be seen from Fig. 3, where there are only two outer blocks and each inner loop optimizes the network in 50 iterations.
>
>
> - "The submission lacks clarity on the assumptions entailed by using the proposed methodology, namely that the Kronecker factors are assumed positive definite."
> Please check the constraints in the optimization problems in Eq. (5) and (10), which have explicitly made it clear that both are positive definite.

---

> ### Author Response · Authors · 2018-11-08
> **Some clarifications - Major comments - PartII**
>
> - "The submission makes heavy use of Kronecker factorization, but neglects to cite works that use a similar factorization of the covariance matrices for neural network applications..."
> 1.  "https://arxiv.org/abs/1503.05671". We would like to point out this work does NOT use Kronecker factorization of the covariance matrices. Instead, the main idea here is to approximate the Hessian matrix (which is also the Fisher information matrix when optimizing log-likelihoods) using Kronecker factorization to approximate the curvature of the loss function in second-order optimization methods. While in our work the Kronecker factorization is enforced on the covariance matrices of the matrix-variate prior distribution.
>
> 2.  "https://arxiv.org/abs/1712.02390". Again, the Kronecker factorization is used to approximate the Hessian in this paper as well, not the covariance matrices.
>
> 3.  "https://arxiv.org/abs/1506.02117". There are three main differences here. First, we focus on the Empirical Bayes method with matrix-variate normal prior, while in this work it is the tensor normal distribution that is used for multitask learning. Second, the optimization formulation is different: while we explicitly take into account the positive definite constraint in (5) and (10), the one in this work needs to have a fudge factor (see Eq. (11)) to avoid the pathological case encountered during optimization. Third, and perhaps most importantly, we propose a block coordinate descent algorithm that has analytic closed form solution to optimize both covariance matrices, while the optimization alg. used in the paper is purely gradient descent for all the parameters.
>
>
> - "Results are reported on a simple regression task (SARCOS) and multiclass classification problems (MNIST & CIFAR10) using a neural network with a single hidden layer."
> We politely disagree with the reviewer. Both networks in both experiments contain multiple hidden layers, and we also extend the approach so that it could be applied into convolution layers as well.
>
> - "Moreover, "in all the experiments, the AEB algorithm is performed on the softmax layer..."
> We would like to point out that we also did experiments with all the layers, and results are discussed in the second paragraph of the ablation study section. We also extend our approach to convolutional layers, see the first paragraph of the ablation study and Fig. 4 for more details.
>
> - Figure 5 does not really exhibit interesting learned structure in the correlation matrix. Why not plot a visualization of the learned prior, rather than the weights?
> Due to space constraint, we put the figures for learned priors in Fig. 7 in the appendix, please have a check and compare them with the ones in Fig. 5. Both of them discover the similar patterns.

---

> ### Author Response · Authors · 2018-11-08
> **Some clarifications - Minor comments**
>
> Minor comments:
> - abstract: "Learning deep neural networks could be understood as the combination of representation learning and learning halfspaces." This is unclear.
> Basically one can understand NN as a stack of nonlinear transformations followed by a softmax layer (for classification, corresponds to linear halfspaces) or a linear output (for regression, but again, this is a sequence of linear halfspaces).
>
> - pg. 2: "Empirically, we show that the proposed method helps the network converge to better local optima that also generalize better..."  What is a "better" local optimum?
> Please check Fig. 3, where we show that the method leads to better convergence in terms of both training and test error.
>
> - pg. 5: "Alg. 1 terminates when a stationary point is found." What exactly is the stopping criterion?
> As standard in nonconvex optimization literature, we use the norm of the gradient as a criterion.
>
> - pg. 6: The labels in Figure 2 are extremely small. Moreover, please keep the y-axis range constant.
> The y-axis are displayed in different scales for different datasets for better visualization.
>
> - pg. 6, Figure 2 caption: "AEB improves generalization under both minibatch settings and is most beneficial when training set is small." Do the CIFAR10 results not show the opposite effect, that the regularization is most beneficial when the training set is large?
> To be fair, our method is better in both ranges on CIFAR10, see Fig. 2.
>
> - pg. 8: "One future direction is to develop a better approximate solution to optimize the two covariance matrices from the marginal log-likelihood function." This is unclear.
> This means to replace the point estimate with better approximation.
>
> We hope the above responses help to clarify some of the misconceptions about our work, and we look forward to your revaluation.

---

> ### Author Response · Authors · 2018-11-22
> **Response**
>
> We thank the reviewer for the feedback. We are glad the reviewer found the proposed empirical Bayes framework useful and the paper easy to follow. Responses to the reviewer’s concerns are addressed below. With these, we hope the reviewer will find the paper more appropriate for publication and, if so, will raise their score accordingly.  We are always happy to discuss further if the reviewer has additional concerns.

---

### Official Review · AnonReviewer3 · 2018-11-10
**Empirical Bayes ideas in neural networks is a great idea. This paper is too heuristic, but the results are encouraging.**

**Rating:** 6
**Confidence:** 4

**Review:**

This paper first proposes an intuition that when training, neural network weights tend to be correlated. It then suggests to make this correlation more specific, and does so using a heuristic algorithm based on approximate empirical Bayes.

The paper is generally well written, in the sense that everything that is done is clearly stated. However why things are done the way they are is less clear.

-	The correlation intuition given in the introduction isn’t quite right (if the training does it anyway, why try to push it even more?). The better way to think about the benefit of correlations is to say that some weak form of weight sharing may help.

-	The approximate notion of empirical Bayes is not well justified.  The arguments given are all about concentration and essentially unimodality. The successive MAP perspective is interesting, but again not clear why we the joint maximization of (4) won’t place us far from the smoothed version of (1).

-	Not only is the AEB principle a heuristic, but the implementation via Algorithm 1 is itself a heuristic to this heuristic. This is because the optimization of the model parameters still remains approximate (due to the local search aspect), even if the optimization of the hyperparameters is shown to be exactly solvable. In the end we end up with a stationary point of the AEB objective, for which we have even less theoretical insight, even if the empirical evidence is promising.

-	A couple of comments about algorithm 1. First, the \theta notation is flipped to refer to the model parameters, instead of the hyperparameters, and it should be fixed. Second, the role of the constant u and v is a bit unclear. On the surface, they provide extra regularization to the hyperparameters. But then, how do we choose them?

-	The experimental results are a bit oversold. First, we have SGD methods that do thrive in smaller batch size regimes, and it’s disingenuous to handicap them with larger batch sizes, when they were performing comparable at smaller ones. It *is* interesting that AED makes the learning batch-size insensitive, and I wish that was elaborated more, to see if it’s a prevalent property in other data sets too. Also, the authors define better local optima is by saying that they reach a lower value of training loss. This is claimed and shown. Usually we think of them as those leading to better generalization. This is claimed, but not shown that these are indeed those that generalize better. Figure 3 (which I assume shows training cross-entropy) shows one outcome among many (the many that are averaged in the other plots), and it could very well be that this outcome did not generalize as well.

Despite these shortcomings, I believe this paper is another welcome push to introducing empirical Bayes ideas into neural networks (though it’s not the first), and the empirical evidence seems to indicate that there is indeed something there to investigate further, so I give it a weak accept.

---

> ### Author Response · Authors · 2018-11-13
> **Response**
>
> Thanks the reviewer for the thoughtful comments. We will answer these questions in this thread.
>
> -   "The correlation intuition given in the introduction isn’t quite right..."
> The correlation intuition during training motivates us to design the proposed Kronecker-factorized covariance matrix, which works as an inductive bias when there is not enough training data to lead to the right level of correlation. This is empirically verified on the MNIST dataset when the training dataset is small.
>
> -   "The approximate notion of empirical Bayes is not well justified..."
> We agree with the reviewer that the argument works when the distribution is concentrated, but we would like to point out that when the training size is large enough, under some regularity conditions, the posterior distribution is indeed concentrated. The problem lies in the unimodality of the likelihood function, which usually does not hold in the case of NNs, but if all the modes (in terms of w) are equivalent, then the maximization of the LHS and RHS of Eq. (2) are equivalent, so this does not affect the choice of \theta.
>
> -   "This is because the optimization of the model parameters still remains approximate..."
> Again, we agree with the reviewer that due to the nonconvexity of NNs, Line 2 of Alg. 1 is not guaranteed to find a global solution, hence the limiting point of Alg. 1 is only a stationary point, from which it is very hard to theoretically analyze the properties of the final estimator. This is indeed one of the main difficulties we encountered during our analysis, and it is not clear to us at this point there is a way to get around of this problem, but it is definitely an interesting problem that deserves further investigation.
>
> -   "A couple of comments about algorithm 1..."
> Thanks for pointing out the conflicting notation! We have fixed this in our updated version of the paper. The role of the constants u and v are used to prevent possible degenerate solutions due to the fact that the positive definite cone is an open set, hence it is possible that the objective function is unbounded. Constants u and v essentially make the domain compact, from which it is guaranteed that the optimal solution exists.
>
> In experiments, we can choose the value by cross-validation. Based on our experience, u = 1e-3 and v = 1e3 should work for the datasets used in this paper.
>
> -   "The experimental results are a bit oversold..."
> This is a great question. We would like to bring the reviewer's attention that in Fig. 3 we also show the cross-entropy loss on the test sets, by which we want to show that these better local optima found during training also generalize better on the test set.

---

### Author Response · Authors · 2018-11-22
**Revision Summary**

We thank all the reviewers for giving valuable comments to this paper. We have revised the manuscript to incorporate the suggestions from the comments.

We highlight the following revisions:

- We have performed additional analysis and conducted more experiments to understand why the empirical Bayes method helps generalization through the lens of stable rank and spectral norm regularization.
- We have provided additional references and discussions to works that use Kronecker factorization to approximate Hessian matrices in second-order optimization.
- We have made further clarifications throughout the manuscript based on reviewers’ comments.

---

### Meta-Review · Area_Chair1 · 2018-12-16
**Novelty is limited and related work is missing.**

**Confidence:** 5
**Recommendation:** Reject

**Metareview:**

This paper proposes a method called approximate empirical Bayes to learn both the weights and hyperparameters. Reviewers have had a mixed feeling about this paper. Reviewers agree that the novelty of this paper is limited since AEB is already a well known method (in fact, iterative conditional modes is a well known algorithm). Unfortunately, the paper completely ignores the huge literature on this topic; the previous reference to use AEB is from McInerney (2017).

Another issue is that the paper seems to be unaware of any issues that this type of approach might have. Here is a reference that discusses some problems with this type of approach:
"Deterministic Latent Variable Models and their Pitfalls"
Max Welling∗ Chaitanya Chemudugunta, Nathan Sutter, 2008

The experiments presented in the paper are interesting, but then are not really doing a good job to assess why the method works well here even though in theory it should not be as good as the exact empirical Bayes method.

This paper does not meet the bar for acceptance at ICLR and therefore I recommend a reject for this paper.